The influence of finfish aquaculture on benthic fish and crustacean assemblages in Fitzgerald Bay, South Australia

Tanner Jason E. 1 2 jason.tanner@sa.gov.au
Williams Kane 1 2
1 SARDI Aquatic Sciences , Henley Beach, SA , Australia
2 School of Biological Sciences, University of Adelaide , SA , Australia
Esteban María Ángeles
Electronic publication date: 2015 Sep 8
Publication date: 2015
Volume: 3
Electronic Location ID: e1238
Received 2015 Apr 27; Accepted 2015 Aug 22
Copyright: © 2015 Tanner and Williams
Copyright year: 2015
Copyright holder: Tanner and Williams
License: This is an open access article distributed under the terms of the Creative Commons Attribution License, which permits unrestricted use, distribution, reproduction and adaptation in any medium and for any purpose provided that it is properly attributed. For attribution, the original author(s), title, publication source (PeerJ) and either DOI or URL of the article must be cited.
License URL: https://creativecommons.org/licenses/by/4.0/

Keywords: Yellowtail kingfish, Seriola lalandi, Wildfish, Aquaculture impacts, BRUVs, Finfish aquaculture

Funding: PIRSA Aquaculture and the Fisheries Research & Development Corporation #2003/223 This work received funds from PIRSA Aquaculture and the Fisheries Research & Development Corporation (project #2003/223). The funders had no role in study design, data collection and analysis, decision to publish, or preparation of the manuscript.

==============================
The influence of sea-cage aquaculture on wildfish assemblages has received little attention outside of Europe. Sea-cage aquaculture of finfish is a major focus in South Australia, and while the main species farmed is southern bluefin tuna (Thunnus maccoyii), there is also an important yellowtail kingfish (Seriola lalandi) industry. Yellowtail kingfish aquaculture did not appear to have any local or regional effects on demersal assemblages (primarily fish, but also some crustaceans) surveyed by baited remote underwater video (BRUV) in Fitzgerald Bay. We did, however, detect small scale spatial variations in assemblages within the bay. The type of bait used strongly influenced the assemblage recorded, with significantly greater numbers of fish attracted to deployments where sardines were used as the bait to compared to those with no bait. The pelleted feed used by the aquaculture industry was just as attractive as sardines at one site, and intermediate between sardines and no bait at the other. There was significant temporal variability in assemblages at both farm sites and one control site, while the second control site was temporally stable (over the 9 weeks of the study). Overall, the results suggested that aquaculture was having little if any impact on the abundance and assemblage structure of the demersal macrofauna in Fitzgerald Bay.

Introduction

While global production figures are uncertain, it is clear that sea-cage aquaculture of finfish has expanded substantially in recent decades, due to increasing demand for seafood and largely steady production from wild capture fisheries (Halwart, Soto & Arthur, 2007). As a consequence, there has been increased attention on its environmental effects. A range of biological and chemical aspects have been studied, including impacts associated with water column eutrophication, the benthic environment and assemblages, trophic structure and diseases or parasites (e.g., Bayle-Sempere et al., 2013; Fernandes & Tanner, 2008; Kalantzi & Karakassis, 2006; Krkosek et al., 2007; Sara, 2007a; Sara, 2007b; Tanner & Fernandes, 2010). More recently, there has also been an increasing focus on the effects on wildfish assemblages in and around aquaculture lease areas (e.g., Dempster et al., 2002; Dempster et al., 2011; Fernandez-Jover et al., 2011; Ozgul & Angel, 2013; Uglem et al., 2014), although the major focus of this work has been in Europe, and especially the Mediterranean. Whether the conclusions derived from these studies are applicable across a broader geographic range is unclear. In Australia, a small amount of work has been done around a snapper farm on the east coast, which showed an increased abundance and biomass of wildfish compared to controls (Dempster et al., 2004), but the issue has received little detailed investigation.

The largely attractive effect of sea-cages that has been documented is assumed to be due to a combination of factors; habitat provision (Papoutsoglou et al., 1996), increased food availability (Pearson & Black, 2001; Uglem et al., 2014), and possibly chemical attraction to farmed stock (Dempster et al., 2002). Two years after abandonment, wildfish abundance around cages left in place at a fish farm in the Canary Islands had decreased 25-fold, although was still double that at controls (Tuya et al., 2006). This decrease was particularly evident in particulate organic matter feeders, but did not occur for herbivores or benthic macro- and meso-carnivores, suggesting that at least at this site, food availability (pellets) was the primary driver of changes, with habitat provision only playing a small role, although attraction to farmed stock cannot be discounted. The aggregation of wild fish has further environmental and ecological consequences that are poorly understood and vary between locations. Flow-on effects can include consumption of waste feed and faeces that would otherwise accumulate on the seafloor (Dempster et al., 2009; Felsing, Glencross & Telfer, 2005; Papoutsoglou et al., 1996), disease or parasite transfer (Krkosek et al., 2007), changes in local assemblage composition (Machias et al., 2005; Ozgul & Angel, 2013), and altered body condition and reproductive output (Dempster et al., 2011; Fernandez-Jover et al., 2011). If fishing is prohibited, aquaculture sites could function as marine sanctuary zones (Dempster et al., 2002), and enhance local stocks by both increasing reproductive output of wildfish due to increased feed availability (Edgar et al., 2014; Pelc et al., 2010) and providing emigrants to the surrounding environment (Roberts et al., 2001; Russ & Alcala, 2011). Alternatively, aquaculture leases may act as ecological traps (Gates & Gysel, 1978; Gilroy & Sutherland, 2007) if access to large quantities of aquaculture feed and faeces leads to decreases in condition and reproductive output of wildfish, although this appears not to be the case in Norway (Dempster et al., 2011). Where legislative protection from fishing is not afforded, aggregations around sea-cages may be easy targets for fishermen, which may exacerbate the over-exploitation of stocks (Dempster et al., 2004).

Here, we assess whether finfish aquaculture has affected the benthic fish and crustacean assemblages in Fitzgerald Bay, South Australia. The benthic assemblages were sampled by baited remote underwater video (BRUV) and compared on a local scale (between sites—aquaculture vs. no aquaculture) within Fitzgerald Bay, regional scale (with other nearby locations that do not contain finfish aquaculture) and over time to detect any differences attributable to aquaculture. We also test the influence of bait, and bait type, on the assemblages detected using BRUVs. While BRUV surveys typically target fish, they also allow other mobile fauna, such as decapod crustaceans, to be enumerated, and so we include both of these components of the benthic fauna.

During recent decades there has been a gradual shift towards the use of remote techniques to sample environments that are not accessible with traditional diver-conducted surveys, and now these methods are also being used in areas that were formerly sampled exclusively by divers (e.g., Lowry et al., 2012; Willis, Millar & Babcock, 2000). The advantages of remote techniques stem from the fact that they are not subject to the limitations imposed upon divers by factors such as depth, temperature, time and safety requirements. The latter is of particular concern in this study, due to the frequent presence of great white sharks (Carcharodon carcharias) in the region. Many non-destructive remote techniques are ideally suited to sea-cage aquaculture and provide several inherent advantages over traditional diver surveys, as well as the universal benefits of remote techniques mentioned above. Non-destructive remote methods avoid the behavioural modifications induced in fish by the presence of divers (e.g., Cole et al., 2007; Watson et al., 2005), do not harm the species or the habitat sampled, and can provide information on the habitat and species behaviour (Harvey et al., 2013; Watson et al., 2005). Irrespective of technique, however, all surveys have their own biases that vary with habitat, environmental conditions and species being targeted. BRUV has become the standard non-destructive remote technique used for surveying demersal fish assemblages (McLean, Harvey & Meeuwig, 2011; Stobart et al., 2007; Unsworth et al., 2014), and is now also being used for pelagic assemblages (Santana-Garcon, Newman & Harvey, 2014).

Methods

Study area

Fitzgerald Bay is located in northern Spencer Gulf, South Australia (Fig. 1). Sea-cage aquaculture was undertaken within the bay continuously from 1999 to 2010, initially producing snapper (Chrysophrys auratus) but since the early 2000’s exclusively producing yellowtail kingfish (Seriola lalandi). At the time of this study in 2004, there were five 20 hectare lease sites (farms) in Fitzgerald Bay, four of which contained stock (all kingfish), with a combined annual production of approximately 620 tonnes. Production increased to ∼2,000 tonnes per annum shortly after this study, but then declined steeply due to husbandry issues, and after 2010, it was relocated further south in Spencer Gulf. The farms containing fish were distributed along a channel that runs through Fitzgerald Bay, to the west of an offshore sandbank. The channel ranges in depth from 10 to 23 m and experiences substantial tidal flows (up to 39 cm s−1, (Parsons Brinckerhoff & SARDI, 2003)). Current direction is approximately north-south along the channel, alternating every six hours in a semi-diurnal pattern. The two farms chosen for the study were located at either end of the channel, to allow for the selection of suitable control sites (Fig. 1). Control sites were selected to be as similar as possible to each lease in terms of geographic location and water depth, and were at least one kilometre from any farm to minimise as much as possible impacts associated with aquaculture development. The southern lease and control sites were dominated by coarse substrate with numerous macroalgae and sponges, while the northern sites had finer and mostly bare sediment, and there is a continuous narrow coastal fringe of seagrass in shallower depths (less than 6–8 m: Hone et al., 1996; Shepherd, 1974). Further details on the site and production cycle can be found in Tanner & Fernandes (2010).

Figure 1 Map of study location.

Map showing the location of study sites within Spencer Gulf (black boxes = lease sites, open boxes = control sites). Inset shows location of SpencerGulf.

BRUV deployment

Benthic BRUV was chosen as the survey technique. All sampling was undertaken during daylight hours (0800–1700) using two BRUVs. Farm site deployments were made within 5 m of a sea-cage, and at least an hour after the single daily feeding had ceased at that cage. Feeding usually commenced early in the morning, but it could take several hours to complete feeding all the cages on a lease. Control sites were divided into 5 by 5 grids (i.e., 25 cells), cells were randomly chosen and BRUVs were deployed at their midpoint. Successive BRUV deployments were usually made 2–10 min apart, separated by a minimum distance of 200 m, but as much as several kilometres depending upon the weather conditions. Once set, the boat was moved >200 m away from the BRUVs and the motors turned off until retrieval.

Two Amphibico Dive Buddy housings were used with the BRUVs; one containing a Sony Digital Handycam DCR-TRV20E, the other a Sony Network Handycam DCR-TRV950E. Cameras were mounted vertically with a distance of 1 m between the lens and the seafloor. Deployment lengths of 30 min were chosen based on the early arrival times and low species numbers detected in the pilot study, where the maximum number of species (1–4) usually occurred before 20 min recording time had elapsed. Unless otherwise noted, a single small (∼400 g) pack of frozen brined sardines (Sardinops sagax) was used as bait for each deployment. Prior to placement in a bait basket, sardines were thawed and crushed to maximise the bait plume.

BRUVs are considered as passive sampling tools, and do not require any ethics or other approvals in the jurisdiction in which this study was undertaken.

Video analysis

Video footage was viewed with a real-time counter, and analysis commenced from the moment that the BRUV settled on the seafloor. Relative abundance estimates of all mobile fauna were made by recording the maximum number of individuals of a single taxon visible within one frame of footage (MaxN, Ellis & Demartini, 1995). MaxN is a conservative measure of relative abundance because it usually underestimates the true numbers of each species visiting the bait (Cappo, Speare & De’ath, 2004). Using MaxN avoids the problem of recounting the same individual on separate visits to the bait, and has been found to give an accurate estimate of “true” density (Willis, Millar & Babcock, 2000). Due to difficulties with identifying small cryptobenthic fish species from the dorsal view recorded by the BRUVs, these species were grouped into a “benthic” category. The presence of two distinct cohorts of snapper (Chrysophrys auratus) in the surveys allowed separation of the classes for statistical analysis (juvenile <38 cm, adult >38 cm). Some blue swimmer crabs (Portunus armatus) were easily distinguished from others (e.g., male or female, missing claw, markings) and thus each new arrival in the FOV was included in the MaxN count regardless of whether they were all present in one frame of footage.

Statistical analyses

Non-parametric permutational multivariate analysis of variance (PERMANOVA, Anderson, 2001) was used to test for differences in assemblage composition between treatments. The Bray-Curtis similarity was used for all analyses, with 9,999 permutations of residuals under a reduced model. All data were 4th root transformed to downweight the influence of highly abundant species, allowing both rare and common species to contribute to the analysis. Without this transformation, the results would be determined almost entirely by the most abundant species (in this case one taxon which had an order of magnitude greater abundance than others in some components of the study) (Clarke & Green, 1988; Clarke & Warwick, 2001). Pair-wise a posteriori comparisons were made for factors that were found to have a significant effect when required. To visualise the similarities between samples, non-metric multi-dimensional scaling (nMDS) ordination plots were used. A similar approach was taken to analyse Total MaxN (i.e. the sum of MaxN across taxa), except that resemblances were calculated using Euclidean distances and no transformation was applied. All analyses were conducted in Primer v6 with the PERMANOVA+ add-on (Clarke & Gorley, 2006).

Local effects

To detect the local-scale effects of finfish aquaculture, BRUVs were used to survey the benthic mobile fauna present on farm and control sites in Fitzgerald Bay. A three-way orthogonal sampling design was used, with Proximity to Aquaculture (farm vs. control), Location (north vs. south) and Tidal Phase (high vs. low) as fixed factors, and three replicates. Sampling was undertaken in late June 2004.

Regional effects

To determine if broader-scale regional impacts of aquaculture were present, the two Fitzgerald Bay control sites were sampled once again, as were two 20 hectare sites both 28 km to the north (Douglas Point) and 22 km to the south (Cowleds Landing) of Fitzgerald Bay (Fig. 1). Neither of these additional locations have been used for aquaculture. Sites within each Location were positioned to match those in Fitzgerald Bay in terms of water depth, separation and site dimensions (Fig. 1). A total of 36 deployments (6 sites ×6 replicates) were conducted over three days in July 2004. Location was treated as a fixed factor, with Site nested in Location.

Bait effects

To evaluate bait efficacy and the effect that different baits had on the sample composition of BRUV surveys in Fitzgerald Bay, three bait treatments were assessed: crushed sardines (as per previous surveys), extruded aquaculture pellets and a control without bait. Pellets used for daily feeding of yellowtail kingfish by the aquaculture industry in Fitzgerald Bay (9 mm diameter, 9 mm long, 5.8% water content) were sourced directly from the aquaculture operators. The no bait treatment consisted of an empty bait basket. Sampling was undertaken throughout the day on three consecutive days in August–September 2004. Each bait treatment was applied to each of the two farm and two control sites from the first survey (3 baits × 4 sites × 5 replicates = 60 deployments) following the protocols described under BRUV deployment, and in a random order. Strong tides during sampling resulted in the loss of six deployments from the southern sites. Bait Type (sardine vs. pellet vs. no bait), Proximity to Aquaculture (farm vs. control), and Location (north vs. south) were treated as fixed factors in a 3-way experimental design.

Temporal effects

To determine whether the effects of finfish aquaculture varied over time, and to examine the temporal stability of the assemblages within Fitzgerald Bay, a temporal comparison of BRUV samples from all three surveys was undertaken. This analysis involved all data from Fitzgerald Bay where sardines were used as the bait, and thus included three factors: Proximity to Aquaculture (farm vs. control); Time (3 surveys) and Location (north vs. south). As no data were collected from adjacent to cages for the regional comparison, there is an empty cell in this design, so the analysis was repeated without data from this comparison (i.e. with data from only 2 surveys). As the results were qualitatively similar, only the results for the analysis with 3 levels of Time are presented.

Results

The 114 BRUV deployments resulted in a total MaxN of 706 across 17 taxa. Over half of these individuals were carangids (Pseudocaranx wrighti—381), with 121 in the ‘benthic’ category, 68 snapper, 63 blue swimmer crabs, 28 western king prawns (Penaeus latisulcatus), 18 bridled leatherjackets (Acanthaluteres spilomelanurus), 10 Port Jackson sharks (Heterodontus portusjacksoni), 8 sand flathead (Platycephalus bassensis), and 1–2 individuals each of smalltooth flounder (Pseudorhombus jenynsii), three species of unidentified demersal fish, the holothurian Australostichopus mollis, red swimmer crab (Nectocarcinus integrifrons), southern calamary (Sepioteuthis australis) and the starfish Allostichaster polyplax. Full details of taxa recorded in each deployment are provided in the Supplemental Information 1.

Local effects

No local-scale effects of aquaculture were detected on the benthic fish and crustacean assemblages surveyed by BRUV in Fitzgerald Bay (PERMANOVA: F1,15 = 0.55, P = 0.63). There was a clear difference between north and south in the bay, however (PERMANOVA: F1,15 = 13.95, P < 0.001), with the northern area having high numbers of the western king prawn and carangids, while the southern area was dominated by blue swimmer crabs (Fig. 2 and Table 1). Tidal Phase had no influence on the assemblage (PERMANOVA: F1,15 = 1.22, P = 0.36), and there were no interactions between any factors (all P > 0.18). No factor (or interaction) had a significant effect on TotalMaxN (all P > 0.16), with the mean value being 7.9 ± 1.2 (se).

Figure 2 MDS plot of local effects of aquaculture.

Non-metric multidimensional scaling plot showing the influence of Proximity to Aquaculture (▴ = lease, ▾ = control) and Location (blue = north, vermillion = south) on benthic fish and crustacean assemblages in Fitzgerald Bay (stress = 0.14). Biplot shows correlations with key taxa (r > 0.4 labelled), with the circle scaled to r = 1.

Table 1 Summary of taxa recorded during local effects survey.

Taxa detected during study of local effects of aquaculture in Fitzgerald Bay. Data are means with standard error in brackets.

Proximity	Location	Blue crab	Western king prawn	Benthic	Carangid	Port Jackson shark	Snapper	
Control	North	0	2.0 (0.26)	2.3 (0.42)	3.8 (1.72)	0	0.2 (0.17)	
Control	South	1.5 (0.50)	0	2.7 (0.88)	0	0.3 (0.21)	0.8 (0.83)	
Lease	North	0	1.3 (0.49)	1.2 (0.40)	2.7 (2.29)	0	1.7 (1.67)	
Lease	South	2.5 (0.22)	1.3 (0.61)	3.7 (0.33)	3.5 (2.43)	0.3 (0.21)	0	

Regional effects

No differences in assemblage structure (Table 2) were detected between the three locations (PERMANOVA: F2,3 = 0.50, P = 0.93), although there were significant differences between Sites within Locations (PERMANOVA: F3,30 = 6.35, P < 0.001). Similar results were obtained for TotalMaxN (PERMANOVA: F2,3 = 0.37, P = 0.94 and F3,30 = 8.6, P < 0.001 for Location and Site respectively, mean ± se = 3.7 ± 0.5).

Table 2 Summary of taxa recorded during regional effects survey.

Taxa detected during study of regional effects of aquaculture in Fitzgerald Bay. Data are means with standard error in brackets.

Location	Site	Blue crab	Benthic	Carangid	Snapper	Leatherjacket	Other	
Cowleds Landing	1	0	1.2 (0.54)	0	0	0	1.2 (0.31)	
Cowleds Landing	2	0.2 (0.17)	1.0 (0.26)	1.8 (1.47)	0	0	1.3 (0.56)	
Fitzgerald Bay	3	0	1.2 (0.17)	0.7 (0.49)	0	0	0	
Fitzgerald Bay	4	3.0 (0.63)	1.7 (0.42)	0.2 (0.17)	1.0 (0.52)	0	0.5 (0.22)	
Point Douglas	5	0.8 (0.31)	1.5 (0.34)	0	0.7 (0.33)	2.5 (0.72)	0.5 (0.50)	
Point Douglas	6	0	0.7 (0.33)	0	0	0	0.5 (0.22)	

Bait effects

In the bait effects study, assemblage structure was influenced by interactions between Proximity to Aquaculture and both Bait Type and Location in bay (Table 3). Pairwise tests indicated that the south control site had a different assemblage to the other 3 sites (P < 0.007). This site had high numbers of juvenile snapper and blue swimmer crabs in comparison to the other sites (Fig. 3). At the farm sites, deployments with bait differed from those without (P = 0.002), but there was no difference between using sardines or aquaculture pellets (P = 0.58). At the control sites, sardines differed from no bait (P = 0.018), but pellets did not differ to either sardines (P = 0.57) or no bait (P = 0.2). Deployments with no bait attracted very few (or no) fauna (8 individuals in 16 deployments, 5 in the ‘benthic’ category, compared to 376 across 38 baited deployments; Table 4).

Figure 3 MDS of bait effects.

Non-metric multidimensional scaling plot showing differences in benthic fish and crustacean assemblages with Proximity to Aquaculture (▴ = lease, ▾ = control), Location (filled = north, hollow = south) and Bait Type (vermillion = pellets, black = sardines, blue = none) in Fitzgerald Bay (stress = 0.14). Biplot shows correlations with key taxa (r > 0.4 labelled), with the circle scaled to r = 1.

Table 3 PERMANOVA table for bait effects.

PERMANOVA table showing effects of Proximity to Aquaculture cages, Location within Fitzgerald Bay and Bait Type on benthic fish and crustacean assemblages detected using BRUVs.

Source	df	SS	Pseudo-F	P(perm)	
Proximity	1	4,093.8	5.44	0.0035	
Location	1	4,878.6	6.48	0.0005	
Bait	2	11,220	7.45	0.0001	
Proximity × Location	1	2,898.8	3.85	0.0184	
Proximity × Bait	2	3,596.1	2.39	0.0473	
Location × Bait	2	3,351.2	2.23	0.0637	
Proximity × Location × Bait	2	3,173.2	2.11	0.0779	
Residual	42	31,615			

Table 4 Summary of taxa recorded during bait effects experiment.

Taxa detected during study of bait effects on assemblages surveyed by BRUV in Fitzgerald Bay. Data are means with standard error in brackets.

Proximity	Location	Bait	Blue crab	Benthic	Carangid	Adult snapper	Juvenile snapper	Leatherjacket	
Control	North	Pellets	0	0.4 (0.24)	7.0 (4.36)	0	0	0	
Control	North	None	0	0.6 (0.40)	0	0	0	0	
Control	North	Sardine	0	0.2 (0.20)	11.0 (4.00)	0	0	0	
Control	South	Pellets	0.8 (0.58)	0.2 (0.20)	1.0 (1.00)	0	1.6 (1.36)	0	
Control	South	None	0.3 (0.33)	0.7 (0.33)	0	0	0	0.3 (0.33)	
Control	South	Sardine	2.7 (0.33)	0.7 (0.33)	0	0	4.0 (3.06)	0	
Lease	North	Pellets	0	0.2 (0.20)	4.0 (1.87)	2.4 (1.17)	0	0	
Lease	North	None	0	0.2 (0.20)	0	0	0	0	
Lease	North	Sardine	0	0.6 (0.40)	30.0 (10.00)	1.4 (0.75)	0	0	
Lease	South	Pellets	0	1.2 (0.37)	33.0 (6.63)	0	0	0	
Lease	South	None	0	0	0	0	0	0	
Lease	South	Sardine	0.4 (0.40)	0.6 (0.60)	9.0 (4.58)	0	1.2 (0.97)	0.4 (0.24)	

TotalMaxN was significantly affected by the interaction between Proximity, Location and Bait type (PERMANOVA: F2,42 = 7.03, P = 0.003; Table 4). Pairwise tests showed deployments with pellets at the south farm site attracted ten times the abundance of benthic fish and crustaceans as at the associated control site (P = 0.008), and five times the abundance as at the northern farm site (P = 0.009). At the north farm site, sardines attracted five times as many animals as pellets, and 150 times as many as unbaited deployments, while at the south farm site, pellets attracted three times as many as sardines, while unbaited deployments attracted no fauna.

Temporal effects

The temporal comparison again showed complicated interactions for assemblage structure (Table 5). Pairwise tests showed temporally variable assemblages at both farm sites (south: P = 0.023; north: P = 0.011), and for the north control site (P ≤ 0.011 for all pairs of Time). Western king prawns were only present in the first survey, while the final survey documented high numbers of carangids and low numbers in the ‘benthic’ category. In contrast, the south control site was temporally stable (P ≥ 0.18), with consistently high numbers of blue swimmer crabs, Port Jackson sharks and the ‘benthic’ category (Fig. 4).

Figure 4 MDS of temporal effects.

Non-metric multidimensional scaling plot showing differences in benthic fish and crustacean assemblages with Time (vermillion = Time 1, black = Time 2, blue = Time 3), Proximity to Aquaculture (▴ = lease, ▾ = control) and Location (filled = north, hollow = south) in Fitzgerald Bay (stress = 0.2). Biplot shows correlations with key taxa (r > 0.4 labelled), with the circle scaled to r = 1.

Table 5 PERMANOVA of temporal differences.

PERMANOVA table showing effects of Time, Proximity to Aquaculture cages, and Location within Fitzgerald Bay on benthic fish and crustacean assemblages detected using BRUVs.

Source	df	SS	Pseudo-F	P(perm)	
Time	2	11,597	9.55	0.0001	
Proximity	1	1,006.7	1.66	0.2284	
Location	1	9,867.1	16.25	0.0001	
Time × Proximity	1	832.38	1.37	0.298	
Time × Location	2	2,541	2.09	0.1045	
Proximity × Location	1	3,157.5	5.20	0.0098	
Time × Proximity × Location	1	3,147	5.18	0.0069	
Residual	44	26,725			

For TotalMaxN, the interaction between Time, Proximity and Location was significant (PERMANOVA: F1,44 = 4.5, P = 0.031, see Tables 1, 2 and 4). Importantly, pairwise tests showed that farm sites did not differ from control sites at each time and location. At the north farm site, there were three times as many fauna at the final census as at the first, while at the control site, the first and final census had four and six times as many fauna respectively as the intermediate census. During the intermediate survey, south control sites had more than three times the abundance as north control sites.

Discussion

Effects of aquaculture

The presence of finfish aquaculture was found to have no effect on the composition of the benthic fish and crustacean assemblages surveyed by BRUV in Fitzgerald Bay on a local or regional scale, although we did detect small-scale spatial and temporal variation in assemblages unrelated to aquaculture. This finding contrasts to most studies that have examined wildfish assemblages around aquaculture cages, which have shown altered community composition, and increased abundance and biomass, as a result of aquaculture (e.g., Dempster et al., 2005; Dempster et al., 2004; Dempster et al., 2002; Dempster et al., 2009; Giannoulaki et al., 2005; Ozgul & Angel, 2013; Valle et al., 2007). Machias et al. (2004) and Machias et al. (2005) also showed regional scale increases in wildfish abundance as a result of aquaculture due primarily to an increase in predators on benthic invertebrates and small fish (i.e. not species likely to feed directly on aquaculture waste). This general increase in fish abundance around farms appears to be method independent, with the studies mentioned above using techniques as varied as diver surveys, trawls, remote video and acoustic surveys, although none have used baited video as we did. While these studies primarily focused on pelagic assemblages directly associated with the cages, or included both pelagic and demersal assemblages, Bacher, Gordoa & Sagué (2012) explicitly examined benthic fish assemblages by diver survey at a farm in Spain and also found them to differ with proximity to cages.

The lack of response to aquaculture detected here may be due to the relatively small-scale nature of the industry in Fitzgerald Bay, which was still expanding at the time of this study, and/or the wide dispersal of wastes, both of which would limit the availability of aquaculture derived food. With an annual production in Fitzgerald Bay of 620 tonnes across four farms at the time of the study, and a food conversion ratio of ∼3:1 (Fernandes & Tanner, 2008), feed input was ∼1,860 tonnes year−1. This was sufficient to produce detectable effects on sediment organic carbon and porewater nutrient levels, but did not produce a clear effect on either infauna or epifauna (Tanner & Fernandes, 2010). Production in Fitzgerald Bay is at the low end of the range for the studies above that have reported impacts of aquaculture on wildfish assemblages (125–3,000 tonnes for those that provided details), although none of these studies report total production for a region, instead only reporting production for individual farms. Now that yellowtail kingfish production is expanding again in South Australia, there is the potential for farming to resume at Fitzgerald Bay. The data presented here, and by Tanner & Fernandes (2010), suggest that at similarly low levels, there would be minimal ecological impact. However, the risk of impacts would increase if production were to expand to typical commercially viable levels seen elsewhere in the world (i.e., several thousand tonnes per annum).

Given the substantial tidal flows through Fitzgerald Bay (up to 39.1 cm s−1, Parsons Brinckerhoff & SARDI, 2003) and the seafloor clearance (5–15 m) of the sea-cage nets, there is ample opportunity for waste dispersal to occur over a substantial area, especially for light-weight wastes (faeces). Conversely, pelleted feed sinks rapidly and is not carried far from the farm, although the accumulation of pellets underneath farms has not been seen (J Tanner, pers. obs., 2004, 2005), and feed wastage appears to be limited (Fernandes & Tanner, 2008).

The combination of these factors may prevent sufficient waste deposition beneath the sea-cages in Fitzgerald Bay to attract resident demersal scavengers. Furthermore, during the bait effects study, pellets held in bait baskets were observed to disintegrate within the 30 min duration of a BRUV deployment. Any pellets, therefore, that did reach the seafloor would most likely disintegrate rapidly and either be consumed by the resident demersal fauna or dispersed by the tide within a very short time. Such limited food availability would provide little direct incentive for scavengers to accumulate in the area.

If the scavengers most involved in waste mitigation in Fitzgerald Bay did not remain associated with the sea-cages for long periods, they may not have been sampled by the techniques used in this survey, as feeding times were avoided during sampling. Wild species have been observed to modify their behaviour in response to aquaculture practices. Sea birds follow feed boats from cage to cage and wild fish follow inter-tidal oyster farmers during infrastructure defouling (K Williams, pers. obs., 2000–2003). It is possible, therefore, that the scavengers in Fitzgerald Bay may also have modified their behaviour. Regardless of the cue (e.g., boat engines, the noise of pellets hitting the surface of the water, the feeding activity of farmed fish), the scavengers may have moved from cage to cage during feeding and thus were not observed in the BRUV deployments. Such movements are a distinct possibility for highly mobile species such as carangids, which were the most abundant species in this study. It is also possible that fish attracted by the presence of aquaculture remain tightly associated with the cages, and were not attracted to nearby BRUVs. Several attempts were made to survey such assemblages with various video deployments, but were unsuccessful, possibly due to limited ability to control which direction the camera pointed. In this respect, a camera allowing greater control, such as used by Dempster et al. (2009) may prove more successful.

Bait effects

While there were complex interactions in the bait effects study, deployments without bait clearly documented a different assemblage to those with bait. The low numbers of fauna documented in the former suggests that unbaited videos had no attractant effect, but rather simply recorded those animals that happened to pass through. That the use of bait increases the abundance and diversity of the fish assemblage recorded is well documented (e.g., Bernard & Goetz, 2012; Hardinge et al., 2013), although a detailed analysis of feeding guilds across a range of habitats showed that this attractant effect only held for predatory and scavenging species, and not for herbivores or omnivores (Harvey et al., 2007).

Sardines and pellets appeared equally effective as bait, at least in terms of assemblage composition. While sardines are the standard bait used for BRUV deployments in Australia, previous work has also shown that other bait types can be equally as effective when it comes to documenting assemblage composition (Dorman, Harvey & Newman, 2012; Wraith et al., 2013). However, both of these studies did find differences between bait types on univariate measures such as total abundance.

Temporal stability

Dempster et al. (2002) and Dempster et al. (2004), found that wild fish aggregations associated with sea-cages in the Mediterranean were relatively temporally stable over periods ranging from several weeks to months. Bacher, Gordoa & Sagué (2012) found a similar result for benthic fish assemblages, but not mid-water and surface, which varied with season. The benthic fish and crustacean assemblages in Fitzgerald Bay also varied over the course of the present study (nine weeks) at both lease sites and one of the control sites. This difference could be due to the fact that this study was essentially sampling natural communities, whereas the aggregations examined by Dempster et al. (2002) and Dempster et al. (2004) were not present prior to the establishment of aquaculture. The differences detected in the present study were possibly due to seasonality; with species responding to the transition from early (June) to late (August–September) winter.

While some species were detected throughout the present study (blue swimmer crabs, carangids, juvenile snapper, “Benthic” category), there were several interesting temporal trends for other species. Mature snapper, western king prawns, Port Jackson sharks and bridled leatherjackets were recorded exclusively during one sampling period. Very low individual counts and sporadic sightings of the latter two species prevent temporal inferences from being made from the existing data. Western king prawns, however, were common during the first survey (June) and absent from the third survey (August-September). Activity in this species is directly related to water temperature, with minimum activity occurring during the cooler winter months (King, 1977). During August-September, water temperatures in Fitzgerald Bay can drop down to ∼13 °C compared to maximum summer temperatures of ∼24 °C (Parsons Brinckerhoff & SARDI, 2003). The lower limit of activity for penaeid prawns is 10–12 °C; therefore, most were likely to have been buried in the sediment during the third survey (King, 1977). The species is also migratory, with individuals moving in a southerly and easterly direction as they mature (Carrick, 1982) and thus likely to leave Fitzgerald Bay during the year. Adult snapper were recorded only during the second survey, which corresponds with the lead-up to their annual reproductive season in upper Spencer Gulf from October to March (Fowler & Jennings, 2003).

Conclusions

Finfish aquaculture in Fitzgerald Bay does not appear to have affected the resident assemblage of benthic fish and crustaceans, as we did not detect any local or regional scale effects on these assemblages in BRUV surveys, despite finding natural small-scale spatial and temporal variation unrelated to aquaculture. Similarly, a concurrent study of other components of the ecosystem in Fitzgerald Bay, which showed detectable impacts on sediment chemistry, did not find effects on infaunal and epifaunal assemblages (Tanner & Fernandes, 2010). Together, these studies suggest that the benthic ecology within the bay is not being substantially affected by waste from the sea-cages. This finding contrasts with most previous work of a similar nature, which may be explained by the relatively low total aquaculture production in Fitzgerald Bay, and high rates of water movement.

Supplemental Information

Supplemental Information 1 Tanner & Williams raw data

Raw data files underpinning the analyses described.

Click here for additional data file.

The cooperation of South Australian Aquaculture Management (SAAM) during the field-work component of the project was greatly appreciated. We would like to thank I Magraith, S De Jong, N Chigwidden and M Ucinek for their invaluable assistance, and S Bryars, M Deveney and S Madigan for advice. Two anonymous reviewers and M Costello provided valuable comments on earlier versions of the manuscript.

Additional Information and Declarations

Competing Interests

Author Contributions

Animal Ethics

JE Tanner is employed by the South Australian Government (SARDI Aquatic Sciences). K Williams is currently self-employed, and was a student at SARDI and the University of Adelaide when the field component of the work described here was undertaken.

Jason E. Tanner conceived and designed the experiments, analyzed the data, contributed reagents/materials/analysis tools, wrote the paper, prepared figures and/or tables, reviewed drafts of the paper, sourced funding.

Kane Williams conceived and designed the experiments, performed the experiments, analyzed the data, contributed reagents/materials/analysis tools, wrote the paper, reviewed drafts of the paper.

The following information was supplied relating to ethical approvals (i.e., approving body and any reference numbers):

BRUVs are considered as passive sampling tools, and do not require any ethics or other approvals in the jurisdiction in which this study was undertaken.

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
