# Peer review of "The influence of finfish aquaculture on benthic fish and crustacean assemblages in Fitzgerald Bay, South Australia"

_PeerJ, doi:10.7717/peerj.1238_

## Round 0.1 · original submission · Major Revisions

· Academic Editor

Major Revisions

Please consider all the suggestions made by the reviewers.

Reviewer 1 ·

Basic reporting

This paper deals with effects on demersal fish assemblages caused by finfish aquaculture in Fitzgerald Bay, South Australia. This is an important and in many respects understudied aspect of marine finfish culture; especially in Australia. The experimental design and data analysis is strong which is a trait of the first author, Tanner.

The manuscript is well written and easy to understand. There are very little typographical errors (see below).

Experimental design

Experimental design and analysis is excellent.

Validity of the findings

Results are acceptable. It would be useful to amend conclusions/introduce new data as per below (general comments) to give the results more context.

Additional comments

Issues with the manuscript are:
1. The study was performed over 10 years ago, it would be of use if the authors included some data about current production in Fitzgerald Bay to give a recent context.
2. Some more discussion about the fact that Fitzgerald Bay is at the low end of production on a world scale and the lack of effect on fish assemblages. Does this necessarily make the aquaculture industry in Fitzgerald Bay best practice or is it not big enough to make a meaningful and significant difference? Given the differences in production levels I believe it is difficult to make comparisons with other aquaculture industries in other countries and I would advise re-wording to reflect this.
3. There are some typographical errors and inconsistencies.
a. Figures 2, 3 and 5, Trevalley is spelt differently to trevally in text. Trevally is the more commonly accepted term although perhaps for international readers this could be changed to Pseudocaranx or carangid or similar to avoid ambiguity.
b. Line 379-380: Why is H. portusjacksoni and A. spilomelanurus not spelt in full like other species?

·

Basic reporting

The title needs revision because the word demersal could also refer to fish swimming around the cages (which are close to the seabed). Also, the paper is not only about fish.

I suggest a better title might be "Benthic fish and crustaceans near kingfish cages in South Australia".

The Abstract could be improved by clarifying that what was actually measured was crustaceans and fish attracted to a baited video on the seabed.

The sentence that there were 'spatial variations' is meaningless as it lacks context (there always is spatial variation in ecology). 'Temporal variability' is also undefined and suggests a significant seasonal or annual study but it is only some weeks, so I would downplay its significance. Again, there is always some temporal variability in ecology, so it needs more context as to whether it is important.

The term macrofauna is widely used in relation to benthic animals like polychaetes and amphipods that are > 1 mm and < 2 cm. In this study I think the term megafauna is more apt.

The Introduction is well written and covers the literature I am aware of well. Its last sentence is weak and could be omitted.

fig 1 - the symbols are much too small and not clear on the map. Could latitude-longitude and a scale bar be added?

Please avoid use of / throughout as it is ambiguous. It could mean and, or, and/or, divided by, per, etc..

fig 2, 3 - What are stars on MDS plots?
What was stress factor?
Was SIMPROF done - did it find any significant difference between samples? A dendrogram may be clearer to reader than MDS.

fig4 and 6 are not clear; 'proximity' especially. What happens if individual taxa are shown - might differences reflect particular taxa abundance in the area rather than bait attraction?

fig 5 not clear. Symbol colours are not distinct enough. Simple black and hollow symbols would be better on al MDS plots. Indeed I wonder whether the MDS are the best way to show patterns.

Experimental design

Line 110 - saying 'benthic habitat is variable' is no help to the reader (it almost always is). Better to just say what the main habitats are.

Line 113-4 - 1 Km is likely to be beyond the effect of farms on the immediate benthos but is not far for fish to swim and pathogens (e.g. sea lice) to be dispersed. I think tagging studies in Norway found wild fish travel between farm sites. So this sentence is not true as it stands.

So video drops were 5 m from the edge of kingfish cages, and control sites 1 km away.

Why were abundances downweighted by transformation? What would the results be if the actual data were used? Perhaps this could be done and if different it should be reported and if not, just say so.

Line 166 - this is not the correct citation for PRIMER-E I think (check their suggestion).

Line 191 - where exactly were bait treatments placed in relation to cages?

Line 192 - Please provide more information on the pellets - size, water content, hard or soft.

Line 203-4 - please clarify about what is meant by 2 and 3 levels of time.

Validity of the findings

The study could be misquoted as suggesting finfish are not attracted to kingfish cages. That seems unlikely as such associations seem the norm. All farm cages I know of have wild fish around them, and fish tend to be attracted to any floating structures including Fish Aggreagating Devices (FADS). However, these fish stay around the cage net rather than the seabed where they would have no cover. Do these authors have any information on fish around the actual cages? Even visual observation? I think this would help balance the findings.

Certainly, the statement on line 304 is misleading. Did those other studies survey the seabed or actually around the cages? This study only surveyed the seabed.

Similarly line 394 should be re-worded. The benthic environment has not been studied here; only animals attracted to baited video within a few minutes. Neither have demersal fish which may be swimming around the cages.

There is a huge literature on benthic impacts of aquaculture (esp finfish cages). In most cages (except Papoutsolgou et al. which is cited here) there are significant accumulations of faeces and waste food underneath. Lines 307 should also be re-worded to clarify that the present study did not look at benthic impacts, and previous studies did not look at megafauna attracted to baited video - so they cannot be compared as if they used the same methods.

I guess there were no hermit crabs, whelks or starfish about - or perhaps BUV were not there long enough?

Lines 207-9
Could we have a table of the taxa found, how many, where etc. This seems central to the paper. Have species names been checked for accuracy against WoRMS (Boxshall et al. 2015, www.marinespecies.org).

Lines 319+ what was water content of the feed? This is critical in calcutation of FCR. Dry pellets can be just 20% water whereas fish are nearer 80% - so calculations of FCR from food and increase in fish weight needs to acocunt for this.

Line 370 again, saying macrofauna 'varied' is meaningless without context.

Line 371 - did this study use the same methods for the same habitats? If not this should be omitted as it is misleading.

Line 384 - what was temperature during the period. It significantly affects fish activity and appetite, and thus attraction to bait and visibility on video.

Reviewer 3 ·

Basic reporting

Article has been written in English and is acceptable in terms of professional standards of expression. However it is noted that some journals prefer a more formal use of language particularly in the results and findings sections instead of using reference terms such as "former" and "latter" - that the authors have used in this paper. There are a few minor typos that the authors will need to correct - for details please see section "General Comments for the Author".

The first introductory sentence in the abstract is not strictly correct - as the authors had not specified the term "demersal" alongside wildfish, which is the main topic of the paper. The paper focusses on wildfish (including broadly macrofauna) near or close to the bottom of the bay/sea floor, and not wildfish in general, that may be found at various depths of the sea.

Figures presented are clear, of sufficient resolution and appropriately described. Table 2 however, has single alphabet initials such as T, P and L under the column of Source, which implicitly indicate Time, Proximity and Location, respectively. For consistency with Table 1 where the authors meticulously described the source instead of using single alphabets (which by the way is not defined in the paper or elsewhere), it is recommended that the authors replace the single alphabets with the proper description. Further it is good practice not to have the same alphabets (in this case, P) representing two different matters (1 for P-value and the other for Proximity) in the same one table.

Experimental design

It is noted that the authors commented that as the Baited Remote Underwater Video is a passive sampling tool, ethics or other approvals in the jurisdiction (of the study) was not required. From the description of the experimental setup, it is deemed that this investigation is sufficiently rigorous having been designed to take into account variables of location, proximity to sea-cages, bait (form) and time.

Validity of the findings

Authors have successfully concluded on their findings. However please see other section "General Comments for the Author" on clarifications relating to the findings.

Additional comments

1. The authors have addressed an important consideration in their discussion between lines 340 to 349. This explanation gives added "strength" as to possible outcomes relating to why feeding times were avoided during sampling. This insight is much appreciated. No changes needed by the authors for this comment.

2. The authors indicated that feedings times were avoided during sampling in the Discussion section. However there was no mention of this procedure in the Methods section. It is recommended that the authors insert a description in line 123 or thereabouts, to highlight that whilst sampling was undertaken during daylight hours (0800-1700), sampling was not carried out during feedings time (and state the times).

3. Whilst it is acknowledged that in marine-related research, time lapse between experiments and publication of findings exist, I am curious as to the large time difference between the experiments (taken to be during the time of the study in 2004 as identified in line 103) and the intended publication of results and findings in 2015 (i.e. some 11 years later). The large time difference raises questions as to whether or not the findings are truly new, or perhaps has been previously published in some other format, or perhaps unless there has been an explicit embargo on the data. The authors may like to explain briefly to curious readers (such as me), somewhere in the paper, as to the time lapse of more than 1 decade from experimental study to publication, and mention whether the same data has been published elsewhere.

4. The minor typos to fix are:
a. In Figure 5 - please correct "Trevalley" to "Trevally"
b. In line 312, please review whether the term is meant to be "latter" instead of "later"
c. In line 343, please correct "practises" to "practices"

5. For added clarity, authors should review line 399 regarding "low stocking levels". The clarifications/questions that arise are: (a) it is assumed implicitly that the authors are referring to stock within the sea-cages. It will benefit readers for this to be explicitly stated, (b) what is defined as a low stocking level - would 200 yellowtail kingfish in a sea-cage be considered as low stock for the region - readers will benefit from an estimate (e.g. < 200 fish per sea-cage) associated with the term "low stocking levels".

6. The paper and study would benefit from identifying the stocking density in the sea cages between the North and South Lease sites/farms. Obviously if the stocking density is similar (also considering the ages of the kingfish whether the stock is a mixture of juvenile or not) then there is one less variable that require discussion or commenting upon. However if the stocking density is different for both North and South Lease sites/farms, then further discussion is warranted - and it may be that the stocking density - which is an important factor in sea-cage aquaculture, may have an effect on the demersal assemblages.

7. Readers will benefit from understanding the composition of the pellets - whether sardine is a major component of the pellet. If sardines is not a major component of the pellet then the authors may like to discuss this matter - as to why there is no significant difference (see lines 236 to 238). Based on my knowledge work has previously been done on Southern Bluefin Tuna comparing the satiability of sardines (whole baitfish) versus pellets - this work may be used to support (or not) this paper's findings regarding the influence of whole baitfish versus pellet on the demersal assemblages.

8. Can the authors clarify whether deployments were made before feeding times? See lines 123-124. If deployments were made before feeding times, could the deployment time (as distinct from the variable Time (see lines 196 ff. for Temporal effects)) have an impact on the number of macrofauna?

9. Authors to please check line 160 whether 9,999 permutations of residuals is correct.

---

## Round 0.2 · Major Revisions

· Academic Editor

Major Revisions

The paper still needs a deep revision. Please, consider all the suggestions made by the reviewer in any revised version.

·

Basic reporting

There are still too many misleading statements that exaggerate the generality of the study and do not place it fairly within the context of other studies (most of which did not use comparable methods). I have listed examples on the MS.

Experimental design

The data is quite limited but still a useful contribution. There are no consistent spatial, temporal or near-finfish cage correlations found; probably because of the limited sampling. thus the paper should not over-state its significance.

Regarding the analysis there is no reason given for downweighting the abundance of the common species by transformation in the analysis except that other people do it. Thus the analysis is not convincing because it is not true to what was actually counted. Analysis using (1) raw abundance and (2) transforming to presence-only data (the most extreme transformation) may show greater insights if either show significant results.

Validity of the findings

See above.

Additional comments

Many statements need to be qualified by the context of the present and other studies which used different sampling methods. It is a stretch to say this study sampled an 'assemblage' when there was great variation between video observations spatially and temporally. It would be more helpful to readers to make it clear the study was of observations of animals attaracted to BRUV (not really an 'assemblage') and not confuse these findings with other studies which sampled by different methods, in different habitats, and thus different biota. Fig 4 and 6 add nothing to the paper than it not already evident in the MDS plots. A table of all the species recorded in each type of bait and location and date should be part of the paper, not an appendix.

---

## Round 0.3 · accepted · Accept

· Academic Editor

Accept

The manuscript has improved according to the suggestions made by the reviewers.

·

Basic reporting

Satisfactory. The authors should do a final proof read because the journal does not do any copy editing.

Experimental design

satisfactory.

Validity of the findings

The interpretation of the results is much improved and adds to current knowledge on fish farm impacts. However, the fact that significant differences were found between some BRUV samples does not necessarily mean the method is suitable for detecting fish farm impacts (but in theory could be). The reason for the difference between samples is not known and so caution is needed in interpreting what that means.

Maybe if the fish abundance data were actual (and not transformed) the results would have been different.